# Geophysically-based analysis of BTCs and ion exchange processes in soil

Shany Ben Moshe[1], Pauline Kessouri[2], Dana Erlich[1], and Alex Furman[1]

[1]Technion - Israel Institute of Technology, Civil and Environmental Engineering, Haifa 32000, Israel
[2]BRGM, French Geological Survey, 45060 Orleans, France

**Correspondence:** Shany Ben Moshe (Benmoshe.shany@gmail.com)

**Abstract.** Breakthrough curves (BTCs) are a valuable tool for qualitative and quantitative examination of transport patterns in porous media. Although breakthrough (BT) experiments are simple, they often require extensive sampling and multi-component chemical analysis. In this work, we examine spectral induced polarization (SIP) signals measured along a soil column during BT experiments in homogeneous and heterogeneous soil profiles. Soil profiles were equilibrated with an $NaCl$ background solution and then a constant flow of either $CaCl_2$ or $ZnCl_2$ solution was applied. The SIP signature was recorded, and complementary ion analysis was performed on the collected outflow samples. Our results confirm that changes to the pore-water composition, ion exchange processes and profile heterogeneity are detectable by SIP: the real part of the SIP -based BTCs clearly indicated the BT of the non-reactive ions as well as the retarded BT of cations. The imaginary part of the SIP-based curves changed in response to the alteration of ion mobility around the electrical double layer (EDL) and indicated the initiation and the termination of the cation exchange reaction. Finally, both the real and imaginary components of the complex conductivity changed in response to the presence of a coarser textured layer in the heterogeneous profile.

# 1 Introduction

Breakthrough curves (BTCs) are a well accepted, convenient laboratory-scale method for evaluation of solute transport parameters in porous media. Arguably, these are the most important measurements needed for proper design of soil and groundwater remediation plan. Evaluation of advective velocity, dispersion coefficients and retardation factors using BTCs have been shown in multiple studies over the last decades (Cameron and Klute , 1977; Yamaguchi et al. , 1989; Vereecken et al. , 1999). The conventional laboratory setup for breakthrough (BT) experiments involves the injection of an inflow solution through a porous media profile; outflow samples are collected and analyzed for ionic composition. However, this conventional setup has drawbacks. Above all, since samples are taken only from the outflow, the obtained curves represents an integrated transport pattern of the solutes along the soil profile. This means that there is no consideration of soil heterogeneity. Additionally, in many cases, full outflow composition analysis requires the use of several analytical instruments and is time consuming and hence, it is often replaced by outflow electrical conductivity (EC) measurements.

The EC of a solution is a measure of its ability to carry an electrical charge and varies with the number and the type of ions present in the solution (Sawyer and McCarty , 1978). As a result, EC measurements provide a rapid estimate of the total dissolved species in water samples. Shackelford et al. (1999) investigated the factors affecting the applicability of EC-based BTCs as an indicator of chemical equilibrium between the effluent and inflow solution. Their findings showed that the shapes of EC-based BTC is a function of the flow rate, the solute retardation factor (that is primarily related to ion exchange processes), and to some extent the species of cation and anion in the inflow solution as well as the cations initially occupying the exchangeable component of the soil. They concluded that EC-based BTCs offer a simple, practical, and inexpensive method for determining chemical equilibrium in laboratory tests. However, since EC primarily reflects the amount of ions in the sample, it is insensitive to changes in ionic composition of the outflow when its overall ionic strength is not significantly altered. This may be a pitfall when exchange processes between an inflow cation and adsorbed species are involved.

Hydrogeophysics is an emerging field of science that is looking for ways to infer hydrological properties and system states in a minimally invasive, minimally destructive fashion (i.e., using geo-electrical methods) (Binley et al. , 2015). While resistivity methods such as ground penetrating radar (GPR) or electromagnetic induction methods are routinely used for many hydraulic applications, the induced polarization (IP) method is especially sensitive to processes related to the interface between the solid and the liquid phases of soil, and to the porous media's internal geometry. IP and spectral-IP (SIP) are increasingly used for the exploration of the subsurface at all scales. In the field of contaminant hydrology, a significant number of studies attempted to use IP methods to detect contamination in the field scale (Sogade et al. , 2006; Masi et al. , 2015). The geo-electrical signature is known to be a composite signature of the porous media itself, the chemical composition of the aqueous phase, the non-aqueous free phases, if such exist (e.g. air, oil) (Shefer et al. , 2013), exchange processes (Schwartz et al., 2012; Schwartz et al. , 2012b), precipitation (Izumoto et al. , 2020), degradation byproducts (Abdel Aal et al. , 2004, 2009, 2014; Schwartz et

al. , 2014) and the soil microbial population (Abdel Aal et al. , 2010; Mellage et al. , 2018).

Five main mechanisms control the geophysical signature associated with IP in porous media. First is the conduction of charge by the electrolyte (the soil pore-water), which is often described by Archie's law (Archie et al. , 1942) or its derivatives. The electrical double layer (EDL) polarization mechanism is relevant in the low range of frequencies characteristic to IP. The EDL polarization mechanism suggests that the polarization of the EDL in the presence of an external electric field is primarily the result of movement of ions in the Stern layer. This suggests that the polarization is primarily affected by the grain size, grain electrical properties (e.g. site density, cation exchange capacity (CEC)), and the composition of the ions sorbed to the soil grain (Leroy et al. , 2009). The electrolyte conductivity and the EDL polarization are assumed to be the most relevant to this study. Other mechanisms include the membrane polarization, the interfacial (or Maxwell-Wagner) polarization and the electrodic polarization. The membrane polarization mechanism is related to the formation of membrane-like features at bottlenecks between adjacent soil grains. That is, when two grains are at very close proximity so that their electrical layers overlap, ions are excluded and accumulate on either side of the membrane, depending on the ion charge and the electrical field direction. Therefore, membrane polarization is related to pore geometrical factors (unlike the EDL polarization, that is related to grain size), together with the chemical composition of the double layer (Titov et al. , 2002). The interfacial polarization is related to charge accumulation at dielectric interfaces, and is typically relevant only in higher frequencies (e.g. above 1kHz), and the electrodic polarization that is relevant at the presence of metallic bodies (Ishai et al. , 2013).

A handful of studies combined geo-electrical measurements with BT experiments and solute transport modeling. Slater et al. (2009) recorded bulk electrical conductivity along a soil column during a microbially-mediated reduction process. They modeled the effluent fluid electrical conductivity with a classic advective-dispersive transport expression and added a conduction term to account for microbially-induced conduction (due to microbial growth). Mellage et al. (2018) studied the transport of coated iron-oxide nanoparticles (NPs) through a sand column using SIP. In their work, they compared normalized imaginary conductivity values (at a chosen frequency, over time) to simulated NP concentrations obtained based on a 1D transport model. They showed that the imaginary conductivity response presented a good fit to the model prediction.

In this work, we use the sensitivity of SIP to changes in pore-water electrolytic composition and charge storage in order to study transport patterns in relatively simple systems, involving non-reactive and reactive ions (undergoing an exchange process). We aim to demonstrate the ability to accurately indicate the BT of the transported species and to quantify the associated transport parameters based on a simple model calibrated using the real conductivity measurements as an alternative to standard chemical analysis of outflow samples. The imaginary conductivity measurements will be used to assess the scope of the exchange process along the soil column. Further, we aim to show that profile's spatial heterogeneity (that wouldn't be inferred by chemical analysis) is easily detectable by SIP-based BTCs.

## 2 Materials and Methods

### 2.1 Spectral Induced Polarization

In classic IP (time domain), an electrical current is injected into the soil profile through two electrodes, and the potential is measured between two other electrodes, focusing on the potential build-up after the initiation of the current (or its cease). In SIP (the method used here), an alternating current in wide range of frequencies is injected, and the phase and amplitude difference between the injected and induced potential is measured.

The complex conductivity signal can be written as

$$\sigma^* = \frac{1}{\rho^*} = |\sigma^*|exp(i\phi) = \sigma' + i\sigma'' \tag{1}$$

where $\rho^*(\Omega m)$ is the complex electrical resistivity (i.e., the reciprocal of the conductivity), $i^2 = -1$ is the imaginary unit, $\phi(rad)$ is the phase shift, and $\sigma'(S/m)$ and $\sigma''(S/m)$ are the real and imaginary parts of the complex conductivity, respectively. Both the real and the imaginary parts of the complex conductivity are related to grain surface interactions; the real part is also related to the conductivity of the pore-water electrolyte composition. (Grunat et al. , 2013)

$$\sigma^* = (\sigma'_{el} + \sigma'_{surf}) + i\sigma''_{surf} \tag{2}$$

where $\sigma'_{el}(S/m)$ is the electrolyte conductivity, $\sigma'_{surf}(S/m)$ and $\sigma''_{surf}(S/m)$ represent the contribution of surface processes to the real and imaginary parts of the complex conductivity, respectively.

To link between surface processes (specifically, the characteristics of the EDL) during the SIP measurement and the complex conductivity signal, the Nernst-Plank equation should be solved, considering the influence of the external electric field expressed by Ohm's law. The resulting expression connects the mobility of ions in the Stern layer to the complex surface conductivity (Leroy et al. , 2009).

$$\sigma_s^* = \frac{2}{r_0}(\Sigma_s + \Sigma_d) - \frac{2}{r_0}\frac{\Sigma_s}{1 + i\omega\tau_0} \tag{3}$$

where $r_0(m)$ is the grain radius, $\Sigma_s(S)$ is the conductance of the Stern layer ,$\Sigma_d(S)$ represents the contribution of the diffuse layer, $f(Hz)$ is the frequency, $\omega = 2\pi f(rad/s)$ is the angular frequency of the current, $\tau_0 = \frac{r_0^2}{2D^*}$ is the relaxation time constant $(s)$ and $D^*(m^2/s)$ is the diffusion coefficient of the counter ion in the Stern layer.

Both $\Sigma_s$ and $\Sigma_d$ are related to the mobility of ions in the EDL. Under the assumption that only one specie is adsorbed to the mineral surface (i.e. mono-ionic system), and that the electrolyte is composed of N species, the Stern and diffuse layer conductance are given by

$$\Sigma_s = e|z|\beta\Gamma_s \tag{4}$$

$$\Sigma_d = e\Sigma_{j=1}^{N}|z_j|\beta_j\Gamma_{j,d} \tag{5}$$

where $e = 1.6 \times 10^{-19}C$ is the elementary charge, $z$ is the valence of the ion, $\beta(m^2/sV)$ is the ion mobility, $\Gamma_s(1/m^2)$ and $\Gamma_d(1/m^2)$ are the surface site densities of the Stern and diffuse layer respectively.

## 2.2 Flow-through experiments

The laboratory setup included a Polycarbonate column ($3 \ cm$ inner diameter, $32cm$ long) equipped with 6 brass electrodes at equal spacing of $4cm$. The top and the bottom electrodes were used to inject the electrical current into the soil profile (and were fully penetrating the soil column), and the remaining four electrodes were used to measure the potential between each pair. The soil column was connected through the electrodes to a portable SIP device (Ontash and Ermac Inc., NJ) using alligator clips. Inflow solution entered the column through its bottom using a peristaltic pump to create saturated flow (a constant flow of $1.3ml/min$) and outflow fractions were collected (see Fig.1).

Four experiments were performed: (a) $CaCl_2$ transport through a loamy profile, (b) $CaCl_2$ transport through a sandy profile, (c) $ZnCl_2$ transport through a sandy profile and (d) $CaCl_2$ transport through a layered profile that included the same loamy soil used for (a), but had a $3cm$ sand layer in the middle. Throughout the experiments, the SIP signal was recorded at frequencies of $0.1 - 10,000Hz$. The impedance and phase shift data were used to calculate the real and imaginary conductivity. Additional information regarding the soil composition and column packing is in section S1 of the supplementary material.

Prior to each experiment, a $200mg/L \ NaCl$ background solution was injected into the column. The background solution injection proceeded until equilibrium between the inlet and the outlet EC was obtained. Subsequently to the EC stabilization, the inflow background solution was replaced by either $360 \ mg/L \ CaCl_2$ solution or $360 \ mg/L \ ZnCl_2$ solution. The replacement of the background solution by the $CaCl_2$ or $ZnCl_2$ solutions marked the beginning of the experiments ($t = 0$).

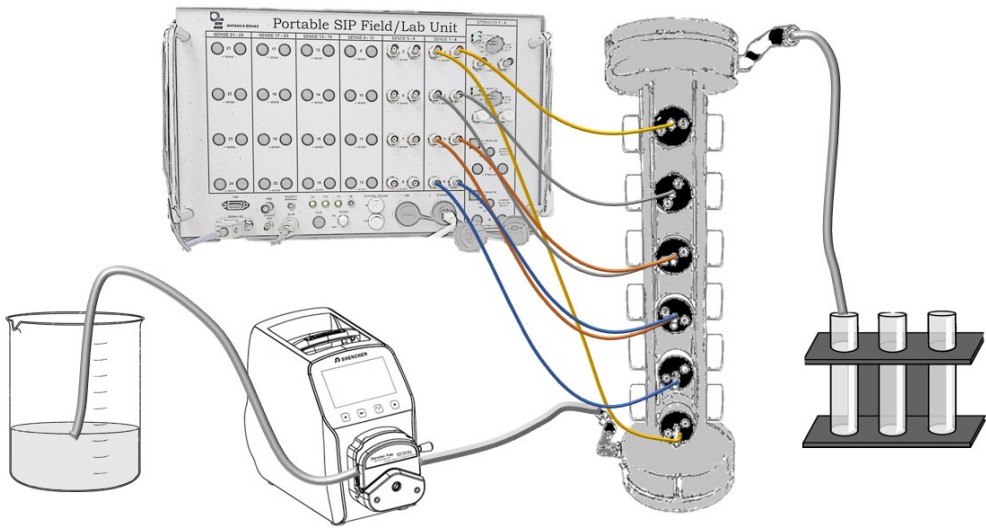

**Figure 1.** Laboratory setup: the inflow solution is pumped into the column from its bottom and the outflow is collected in fractions. SIP signal is measured between three sets of potential electrodes: channel 1 (blue), channel 2 (orange) and channel 3 (grey). The current is injected between the top and bottom electrodes (yellow).

### 2.2.1 Soil preparation and characterization

Red loamy soil (denoted here as 'loamy soil'; used for experiments (a) and (d)) and the sand used for the heterogeneous experiment (d) were air dried, passed through a $< 250 \mu m$ sieve and used without further treatment. The sandy soil that was used for both $CaCl_2$ and $ZnCl_2$ transport (experiments (b) and (c)) was treated to remove its $CaCO_3$ content prior to the column packing. Briefly, the soil was mixed with hydrochloric acid under stirring conditions and then washed with deionized water (DW), three times. Then, it was left overnight in a diluted $NaOH$ solution ($pH = 8$). Finally, the soil was washed and left overnight in an $NaCl$ solution ($0.5\ g/L$) and then air dried and packed. The texture of the loam and the sand was determined using hydrometer experiment (based on Stokes' law; (Bouyoucos , 1962)) and the CEC of the loam, sand and treated sand was measured using the $BaCl_2$ compulsive exchange method (Gillman , 1986).

### 2.3 Complementary analysis

The outflow solution was collected in equal-volume fractions. Immediately after the collection of each sample, its EC was measured using an EC meter (Eutech con 700). Each sample was then passed through a $0.22 \mu m$ cellulose filter and analyzed for $Cl^-$ using an Ion Chromatograph (881 compact IC pro, Methrohm) and for sodium, calcium and zinc by inductively coupled plasma - optical emission spectrometer (ICP-OES; iCAP 6000).

### 2.4 Modeling

A simple solute transport model was constructed based on the HYDRUS 1D platform (Simunek et al. , 1998). HYDRUS 1D is a computer software for numerical modeling of water flow and solute transport in porous media. The inverse tool of HYDRUS 1D was applied to fit model parameters to experimental results, based on the Levenberg-Marquardt nonlinear minimization method (Marquardt , 1963). The appropriate mass balance equation for this application is given by the advection-dispersion equation (ADE)

$$\frac{\partial c}{\partial t} + \frac{\rho_b}{\theta} \frac{\partial \hat{c}}{\partial t} = \frac{\partial}{\partial l}(D_h \frac{\partial c}{\partial l}) - \frac{\partial}{\partial l}(\nu c) \tag{6}$$

where $c(g/cm^3)$ is the dissolved species concentration $t(h)$ is time, $l(cm)$ is the vertical coordinate, $v(cm/h)$ is the water velocity, $D_h = \alpha\nu + D$ is the hydrodynamic dispersion coefficient $(cm^2/h)$, $\alpha(cm)$ is the dispresivity, $D(cm^2/h)$ is the diffusion coefficient, $\rho_b$ is the soil bulk density $(g/cm^3)$ and $\hat{c}$ is adsorbed concentration $(g/g)$ that can be expressed in terms of $c$ using an adsorption isotherm. For simplistic linear adsorption model, $\hat{c}$ can be expressed as

$$\hat{c} = K_d c \tag{7}$$

where $K_d(cm^3/g)$ is the linear adsorption coefficient which is equal to 0 for non-reactive solutes.

Here, we considered saturated flow, the boundary flux was determined based on the measured flow. The dispersivity ($\alpha$) and the soil porosity ($n$) were fitted using the inverse tool for the non-reactive solutes. For the reactive species ($Ca^{2+}$ or $Zn^{2+}$) model fitting process, the obtained values of $n$ and $\alpha$ values were used and the adsorption coefficient $K_d$ was fitted.

### 3 Results and discussion

#### 3.1 SIP spectra and temporal response analysis

The complex conductivity measurements obtained by SIP hold information regarding the soil's pore-water conductivity and its surface polarization in response to the applied external field. Figure 2a and 2b present $\sigma'$ and $\sigma''$ between $0.1$ and $1000Hz$ at five different times for a loamy soil profile during continuous $CaCl_2$ injection. The $CaCl_2$ solution had a higher EC compared to the background solution and hence $\sigma'$ increase accordingly as the inflow solution progressed along the column. The $\sigma''$ spectra present a sharp polarization peak at $\sim 1Hz$ and its magnitude increases and then decreases in response to the introduction of the inflow $CaCl_2$ solution. A relaxation model fit (Pelton et al. , 1979) revealed a mild change in relaxation time between the beginning and the end of the $CaCl_2$ injection (from $0.16s$ to $0.18s$ ;not presented). The inserts in Fig.2a and 2b schematically present the values of the real and imaginary conductivity at $1Hz$ versus time (note the BT pattern received for the $\sigma'$ versus time graph). Figure 2c shows $\sigma''$ between $0.1$ and $1000Hz$ for the three soils used in this work: loamy soil, sand (untreated; used for the heterogeneous experiment) and sand that was treated with acid to remove its $CaCO_3$ content. While all three

spectra present a polarization peak at $\sim 1Hz$, they differ in their magnitude. The loamy soil's $\sigma''$ magnitude is considerably higher compare to the sand along the measured frequency range. This is explained by the larger clay fraction and higher CEC of the loam compared to the sand that have been shown to affect the magnitude of $\sigma''$ (Revil , 2012). The CEC of the three soils is $\sim 5meq/100g$, $\sim 4.15meq/100g$ and $\sim 3.75meq/100g$ for the loam, sand and treated sand, respectively.

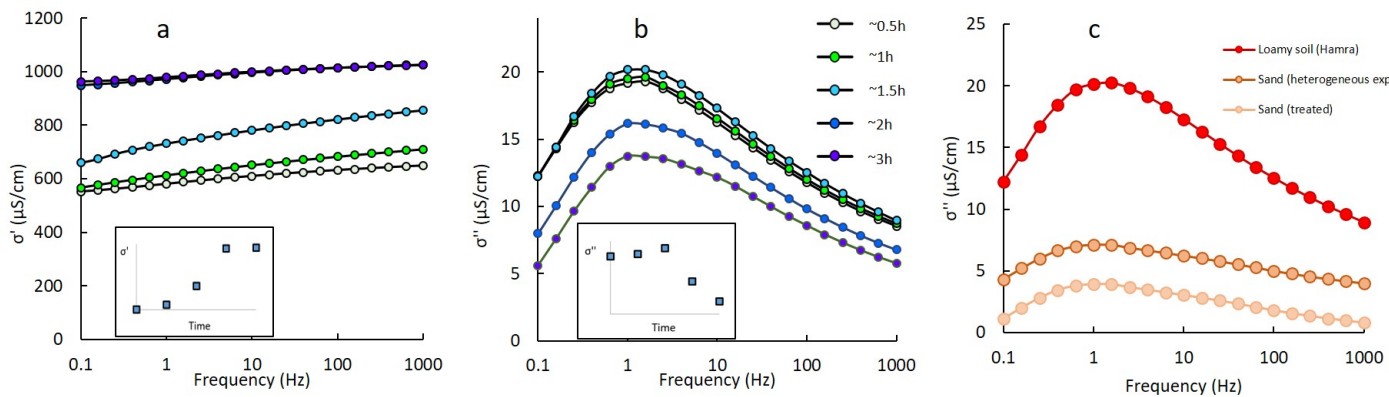

**Figure 2.** Real (a) and imaginary (b) conductivity between 0.1 and $1000Hz$ at five different times ($\sim 0.5h, \sim 1h, \sim 1.5h, \sim 2h, \sim 3h$) for the loamy soil. The inserts in (a) and (b) schematically present the dynamics of the real and imaginary conductivity at $1Hz$. (c) shows the imaginary conductivity of the three soils used in this work: loamy soil, sand (untreated; used for the heterogeneous experiment) and treated sand.

### 3.2   $CaCl_2$ transport and $Na^+ - Ca^{2+}$ exchange in a homogeneous loamy profile

Figure 3a presents the measured outflow $Cl^-$ concentrations and EC values versus pore-volume ($PV(-)$; one pore-volume is equivalent to $\sim$82 minutes). Both the $Cl^-$ and EC outflow values along time fit the expected BT pattern of a non-reactive solute: the ratio of solute concentration ($C$) to its inflow concentration ($C_0$) is equal to $\frac{1}{2}$ approximately after $1PV$ ($PV(\frac{C}{C_0} = \frac{1}{2}) \approx 1$). $Na^+$ and $Ca^{2+}$ dynamics during the $CaCl_2$ injection are described in Fig. 3b. $Na^+$ outflow concentrations increased in parallel to the increase in outflow $Cl^-$. This observation is explained by the fact that the injection of the $CaCl_2$ solution started after an equilibrium between the background solution and the soil was reached. At this point, the $Na^+$ concentration in the pore water was already equal to its concentration in the inflow solution ($\frac{C}{C_0} = 1$) and therefore $Na^+$ behaved as a non- reactive solute. As the $CaCl_2$ solution entered the column, adsorbed $Na^+$ ions were replaced by $Ca^{2+}$ and hence the $Na^+$ outflow concentration increased further. In parallel to the increase in outflow $Na^+$, a moderate increase in outflow $Ca^{2+}$ was observed. This early increase in outflow $Ca^{2+}$ represents the fraction of the inflow $Ca^{2+}$ ions that did not participate in the $Na^+ - Ca^{2+}$ exchange reaction (note the timing at $\sim 1PV$). Subsequently to the completion of the $Na^+ - Ca^{2+}$ exchange, a decrease in outflow $Na^+$ concentrations was observed while outflow $Ca^{2+}$ increased and reached its inflow concentration.

Outflow EC values (Fig.3a) increased in a similar pattern to the $Cl^-$ ions and reached their maximal value after $\sim 1.3PVs$. The major changes in the ionic composition of the outflow during the $Na^+ - Ca^{2+}$ exchange reaction (see Fig.3b) were not re-
flected in the EC measurements. This observation is consistent with the findings of Shackelford et al. (1999); in their study, they combined experimental observations and theoretical approach to investigate EC-based BTCs. They used a sodium-saturated clay soil (sodium bentonite) and a $CaCl_2$ solution as the inflow solution. Their results showed that while the EC-based BTC matched the $Cl^-$ BT pattern, reaching its maximal value after $\sim 1.5PVs$, it stayed stable afterwards and did not reflect the subsequent decrease in outflow $Na^+$ and increase in outflow $Ca^{2+}$, similarly to the presented here.

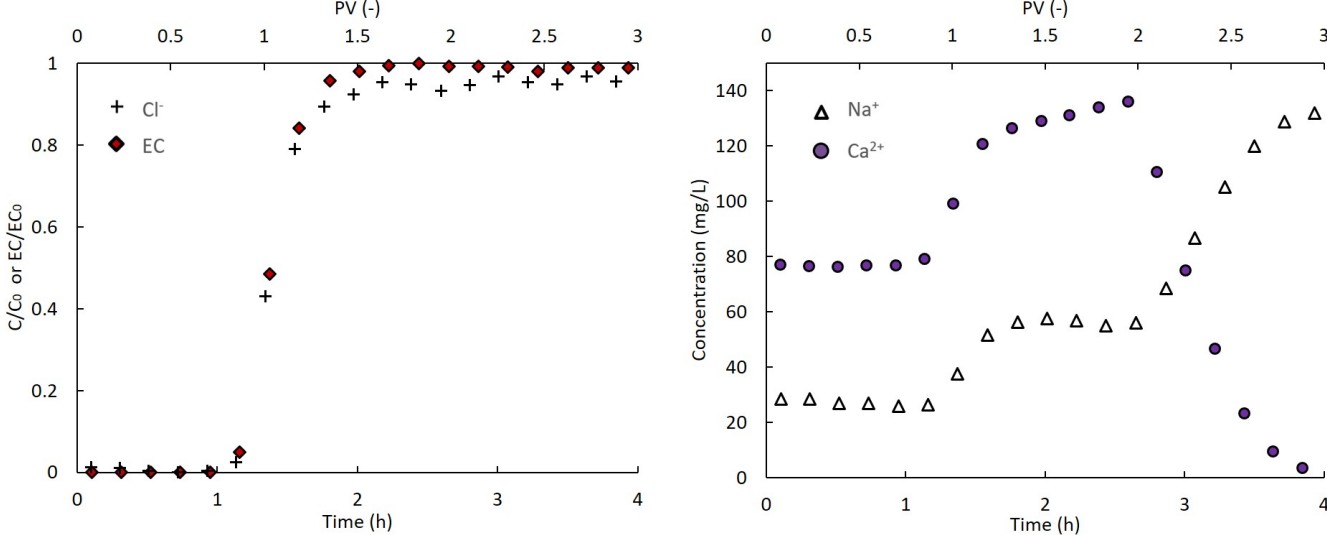

**Figure 3.** Outflow values versus time (bottom horizontal axis) and pore volume (PV; upper horizontal axis) of (a) measured $Cl^-$ and EC ((+) signs and red rhombi), and (b) $Na^+$ and $Ca^{2+}$ ions (empty triangles and purple circles).

Figure 4a presents the normalized real component of the complex conductivity at $1Hz$ versus time at three locations denoted as channel 1, channel 2 and channel 3, for the homogeneous loamy profile. The $\sigma'$ values were normalized by subtraction of the minimal value and division by the difference between the maximal and minimal value (normalized $\sigma' = \frac{\sigma' - \sigma'_{min}}{\sigma'_{max} - \sigma'_{min}}$). The values of the real conductivity along time depicted the general shape of a BTC: a gradual increase in $\sigma'$ was followed by
stabilization of the signal around its maximal value. The $\sigma'$-based BT pattern at the different channels was consistent with the location of the electrode pairs: BT was first observed at channel 1, which is the closest to the inlet, then at channel 2 and lastly at channel 3. Further, the values of $\sigma'$ along time are in agreement with the measured EC values presented in Fig.3a. In all three channels, the signal was initially stable at around $580\,\mu S/cm$ and stabilized on its maximal value ($\sigma'_0$) at $\sim 1055\,\mu S/cm$. Interestingly, a secondary BT pattern was detected in the $\sigma'$-based curves between $0.9$ and $2.5h$. The temporal location of this
BT pattern suggests that the real part of the complex conductivity responded to the increased pore-water $Ca^{2+}$ concentrations after the $Na^+ - Ca^{2+}$ exchange reaction reached equilibrium. This increase, however, occurred simultaneously to the decrease

in $Na^+$ concentrations in the pore-water and hence did not have a major effect on the ionic strength of the outflow and was not detected by the EC measurements (see Fig.3a).

A HYDRUS 1D-based model was constructed and calibrated separately for the transport of the non-reactive ($Cl^-$, $Na^+$ and a small fraction of the $Ca^{2+}$ that reached the outlet of the column at the same time as the non-reactive $Cl^-$ and $Na^+$ ions) and the reactive ($Ca^{2+}$ minus the 'non-reactive fraction') ions in the system. In the model fitting process, the $\sigma'$ time series were used as a proxy for concentration. This approach was used before; Moreno et al. (2015) used electrical resistivity tomography (ERT) to monitor the water dynamics under a drip-irrigated citrus orchard. They presented a coupled flow and transport model

that was calibrated using electrical conductivity measurements with the pore-water electrical conductivity serving as a pseudo-solute.

   Here, the $\sigma'$-based BTCs of all three channels were split into two parts, separating the initial and the secondary BT. Each part was then used separately for the model calibration: the initial BT data was used for the calibration of the non-reactive solute transport and the secondary BT was used for the reactive solute transport. The 'splitting point' of the data for each channel was

220 determined according to the value of the curve's slope between $0.5$ and $1.5h$ (i.e., the data was split at the point of minimal slope). The obtained data sets were normalized to a scale of 0 to 1 and the HYDRUS inverse tool was used to fit the model parameters. The obtained dispersivity and porosity were $0.26cm$ and $47\%$, respectively. For the reactive ion, a $K_d$ value of $2.5cm^3/g$ was obtained.

   Figures 4b and 4c present the normalized $\sigma'$-based time series and their model fit for the non-reactive and reactive ions, respec-

225 tively. Using the obtained model parameters, outflow ion concentrations were predicted (black continuous line) and compared to the measured concentrations (clear rectangles). Both the model fit and the predicted outflow concentrations presents an excellent fit to the measured data. $R^2$ for the non-reactive ions is $0.991$ and $0.972$, respectively and $0.972$ and $0.965$, respectively for the reactive ion. The ability to accurately predict the BT pattern of outflow concentrations using a simple, HYDRUS-based numerical model suggests that the electrical measurement can replace outflow sampling and chemical analysis. Clearly, in more

complex systems that involve multiple ions (either in the inflow solution or adsorbed to the soil), some sampling and chemical analysis may be required in addition to the SIP monitoring. However, even for such systems, the use of electrical monitoring may allow to drastically reduce the sampling frequency and simplify the analysis performed on the outflow samples.

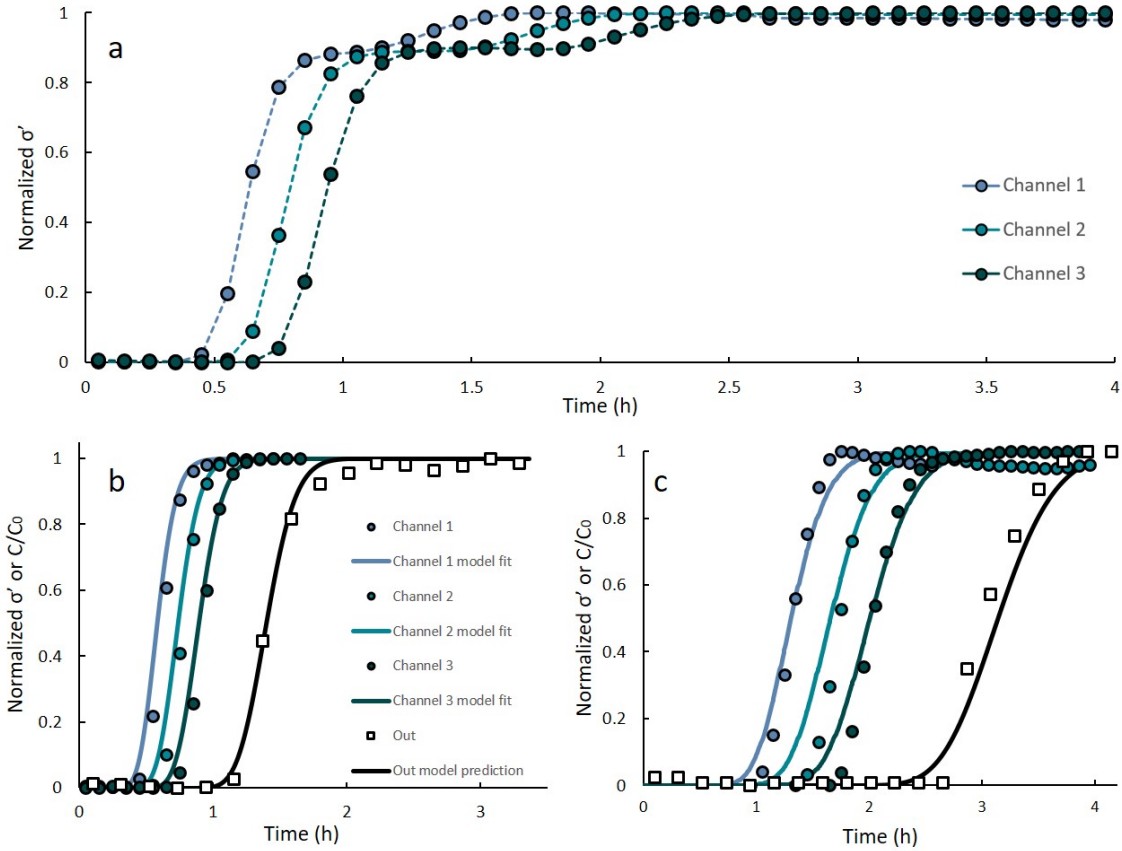

**Figure 4.** (a) Real conductivity at $1Hz$ versus time at three locations, (b) model fit to the $\sigma'$-based initial BT and model prediction for the non-reactive species at the outflow compared to their measured concentration, (c) model fit to the $\sigma'$-based secondary BT and model prediction for the reactive fraction of $Ca^{2+}$ ions (calculated as outflow $Ca^{2+}$ minus its 'non-reactive fraction') compared to its measured concentration.

The imaginary conductivity along time (Fig.5) provides a clear indication for the scope of the $Na^+ - Ca^{2+}$ exchange reac-
235 tion. The signal increased initially in all channels, but decreased and stabilized after reaching its maximal value. The observed decrease in $\sigma''$ that was followed by the BT of $Ca^{2+}$ at the outflow, is related to the altered composition of the Stern layer after the exchange reaction. Imaginary conductivity changes have been shown to be related to the mobility of ions in the Stern layer (Vaudelet et al., 2011; Schwartz et al., 2012). Specifically, adsorbed $Na^+$ ions maintain their hydration shell and hence, are weakly adsorbed to the soil and are more mobile compared to $Ca^{2+}$. During the $Na^+ - Ca^{2+}$ exchange, the less mobile
$Ca^{2+}$ ions occupied the Stern layer and caused the decrease in $\sigma''$. These observations are consistent with earlier studies. Shefer (2015) examined the SIP signature of ion exchange processes in loam and loess soils. the soil profiles were treated with high concentrations of $NaCl$ or $CaCl_2$ solution to alter the ionic composition of the Stern layer. Her results showed that the imaginary conductivity of the soil profile increased inversely to the mobility of the adsorbed ion. Vaudelet et al. (2011)

observed similar results comparing the SIP signal of a saturated sand profile adsorbed with $Cu^{2+}$ and $Na^+$. The $Cu^{2+}$ ions
are adsorbed to the soil mainly as an inner sphere (less mobile) species and hence the $Na^+ - Cu^{2+}$ exchange reduced the
overall ion mobility in the Stern layer, similarly to the $Na^+ - Ca^{2+}$ exchange considered here.

The increase in $\sigma''$ begun before $PV = 1$ at each of the channels was reached (see dashed vertical lines in Fig. 5) and hence,
might be related to the progression of the non-reactive ions' front and increase in EC during the $CaCl_2$ injection. However,
the imaginary conductivity is only marginally influenced by the conductivity of the pore-water. Moreover, the signal continued
to increase in all channels after the stabilization of the outflow EC. This may imply that the increase in $\sigma''$ is related to the
$Na^+ - Ca^{2+}$ exchange reaction. As $CaCl_2$ entered the column, $Na^+$ ions leave the Stern layer while $Ca^{2+}$ ions enter it.
The SIP results (the observed increase in $\sigma''$) suggest that there is an intermediate stage at which neither $Ca^{2+}$ nor $Na^+$ are
fully adsorbed, which means that their mobility is higher than their mobility at the Stern layer (Eqs.3- 4). Considering a simple
EDL model, this suggests that both the $Ca^{2+}$ and $Na^+$ are still present in the diffuse layer. This suggests that the beginning
of the $\sigma''$ increase indicates the initiation of the exchange reaction at each location, while the steep decrease and following
stabilization of the signal on its minimal value marks the end of the exchange. The secondary BT pattern observed in the $\sigma'$
signal (see Fig.4b) supports our hypothesis: the minimal $\sigma''$ values (indicating the end of the $Na^+ - Ca^{2+}$ exchange) were
observed at the same time as the stabilization of the $\sigma'$ signal on its maximal value (suggesting the presence of $Ca^{2+}$ at its
inflow concentration in the pore-water).

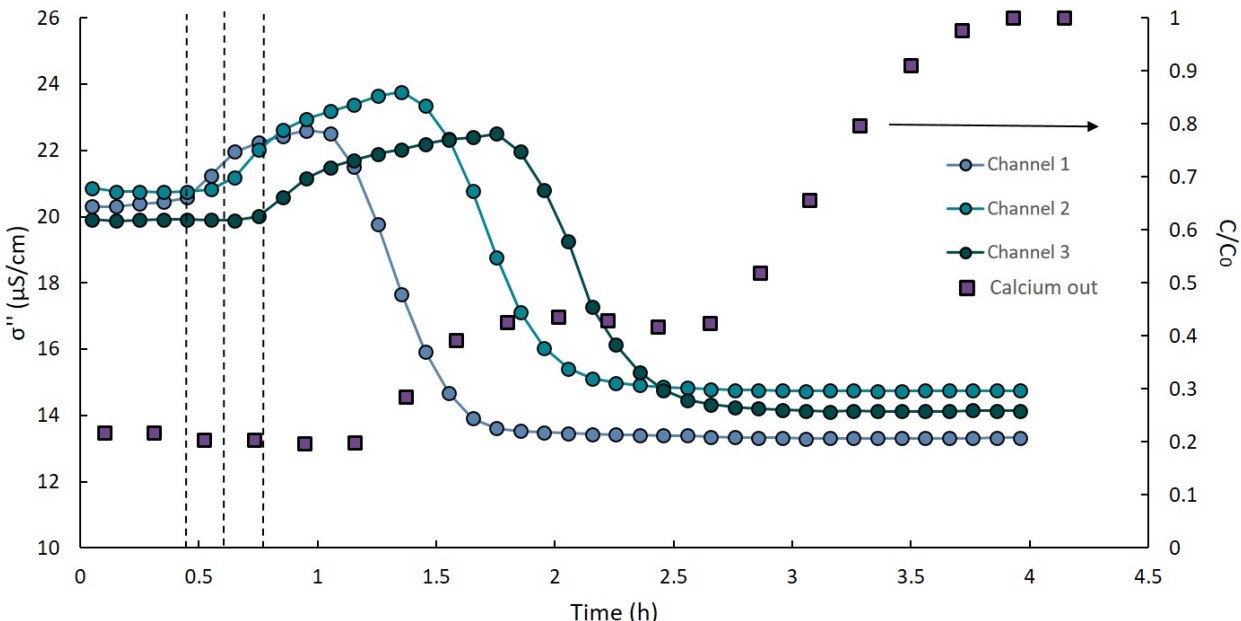

**Figure 5.** Imaginary conductivity at 1Hz versus time at three locations (left vertical axis) and outflow $Ca^{2+}/Ca_0^{2+}$ versus time (right vertical
axis). The vertical dashed lines represent the beginning of the increase in $\sigma''$ at each of the channels.

The role of $\sigma''$ as an indicator of the $Na^+ - Ca^{2+}$ exchange progression suggests that the $\sigma''$ values can serve as a proxy for the adsorbed concentration of the retarded (reactive) species in an alternative modeling approach. A demonstration of this approach and technical description of the model construction and calibration is presented in the supplementary material (section S2). However, the initial increase in $\sigma''$ during the exchange process is not captured by this simple model. Since this increase is assumed to be related to non-equilibrium processes, to fully connect the SIP data to the $Na^+ - Ca^{2+}$ exchange a more complex model that includes these processes should be constructed.

### 3.3  $CaCl_2$ and $ZnCl_2$ transport in a homogeneous sandy profile

Similarly to the observed for the loamy profile, a double BT pattern was observed for the transport of $CaCl_2$ and $ZnCl_2$ in a sandy profile. Figure 6a presents the normalized real conductivity (at 1Hz) recorded in channel 2 versus time, for the $CaCl_2$ and $ZnCl_2$ transport experiments (purple and grey circles). It also shows the measured outflow values of $Ca^{2+}$ and $Zn^{2+}$ as was measured by ICP (purple and grey squares). The initial BT observed between $0$ and $1h$ was nearly identical for both experiments ($CaCl_2$ or $ZnCl_2$ injection). Both experiments were performed on the same soil and started after an equilibrium with the same background solution was reached. Further, the initial BT indicates the movement of the non-reactive ions which are expected to progress at a similar rate in both cases. The secondary BT, however, was slightly earlier for the $ZnCl_2$ case suggesting that calcium is retained for longer in the soil compared to zinc; this is also evident by the measured outflow concentrations. In the pH range characteristic to soil systems, both calcium and zinc are found in the soil in their divalent form. However, in the presence of $Cl^-$, $OH^-$ and the carbonate system's ions, a number of complexes are formed. To explain the delayed BT of the calcium compared to the zinc, PhreeqC simulations were performed (Parkhurst , 1995). The simulations calculated the concentrations of the possible calcium and zinc complexes in the characteristic pH of the system and in equilibrium with atmospheric $CO_2$. It was found that for our system, while calcium was mostly present in its divalent form ($< 97\%$ of its total concentration), around $13\%$ of the zinc was present in a monovalent form (as $ZnHCO_3^+$). The stronger attachment of the $Ca^{2+}$ ions to the soil is also evident by the $\sigma''$ temporal variation presented in Fig.6b. For both experiments, the $\sigma''$ values drop in response to the $Na - Ca$ or $Na - Zn$ exchange. However, the values drop lower in the case of the $Na^+ - Ca^{2+}$ exchange, indicating lower ion mobility in the soil's EDL after the exchange. HYDRUS-based transport models were calibrated for both experiments based on the $\sigma'$ time series and presented an excellent prediction of the outflow concentrations (model fit and prediction for the $ZnCl_2$ is presented in section S3 of the supplementary material). The obtained dispersivity and porosity were $0.25cm$ and $48\%$, respectively. $K_d$ values of $2.13cm^3/g$ and $2.2cm^3/g$ were obtained for the zinc and the calcium, respectively .

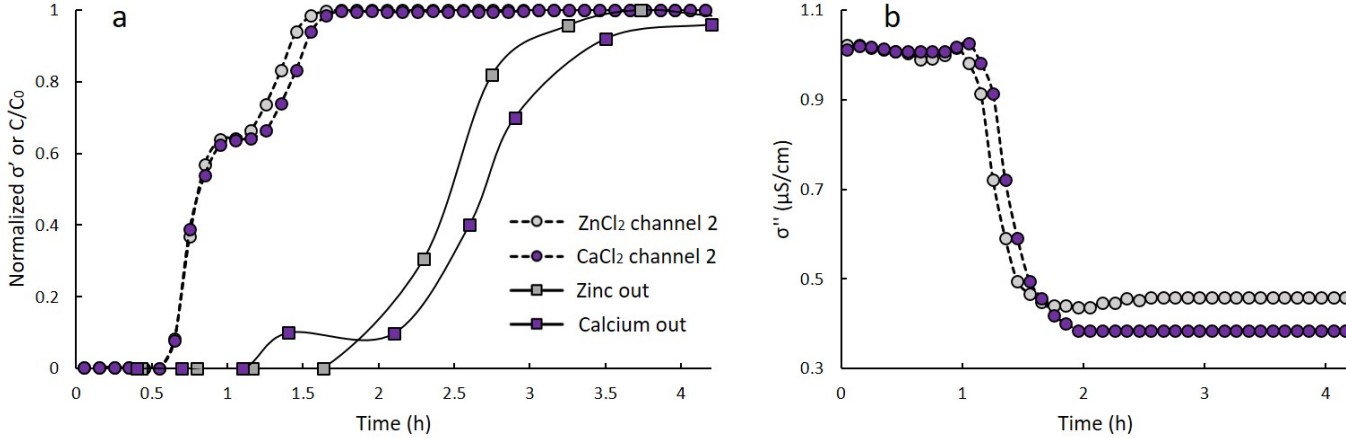

**Figure 6.** (a) Normalized real conductivity (a) and imaginary conductivity (b) versus time at 1Hz versus (channel 2) for $CaCl_2$ and $ZnCl_2$ transport in a homogeneous sandy profile. The rectangular markers symbolize the measured outflow $Zn^{2+}$ and $Ca^{2+}$ concentrations.

### 3.4 $CaCl_2$ transport and $Na^+ - Ca^{2+}$ exchange in a heterogeneous profile

A comparison of $Ca^{2+}$, $Na^+$, $Cl^-$ concentrations and EC values at the outflow samples collected during the heterogeneous and homogeneous experiments revealed no significant difference (t-test, $\alpha$=0.01; see section S4 of the supplementary material). This may be due to a combination of several different reasons: first, the loamy soil used here contained >90% sand, and hence the overall difference in CEC between the profiles was minor. Further, the sand layer was relatively thin at one-tenth the profile's total volume (see Fig. 7c). Since the chemical measurements were conventionally performed on outflow samples,

the obtained transport patterns of the different ions along time represent a spatial average and hence the implications of the heterogeneity were not detected by the chemical analysis.

Figure 7 presents the real (7a) and imaginary (7b) components of the complex conductivity at 1Hz versus time for the heterogeneous profile. Similarly to the homogeneous experiment, the $\sigma'$-based BT pattern observed between $0.5$ and $1.2\ h$ appeared

at the different channels according to the geometrical hierarchy. The general shape of the secondary increase in $\sigma'$ (between $0.9$ and $2.5\ h$) and the imaginary conductivity curves are also consistent with the homogeneous case. However, for both components of the complex conductivity, the signal recorded in the middle channel (channel 2, measuring over the sand portion of the column) was notably lower compared to the other two channels. $\sigma'$ values ranged between $600 - 1055\ \mu S/cm$ in the loam but were between $400 - 600\ \mu S/cm$ in the sandy layer. Similarly, $\sigma''$ values ranged between $15 - 24\ \mu S/cm$ in the loam compared

to $4 - 7\ \mu S/cm$ in the sandy layer. This observation indicates that the SIP-based BTC can characterize system heterogeneity, which is a major advantage over the conventional outflow analysis method.

The sensitivity of the complex conductivity to the geometry of the porous media and its surface charge means that differences in soil grain size, porosity, and CEC are likely to be reflected in the obtained signal. Geo-electrical studies of porous media characteristics mostly focused on physical aspects, rather than chemical aspects. The effect of soil hydraulic characteristics on its SIP signature have been studied before (e.g. (Kruschwitz et al. , 2010; Breede et al. , 2012; Nordsiek et al. , 2016)). For example, Kruschwitz et al. (2010) examined the low-frequency electrical spectra of a range of natural and artificial porous media in the aim to link the electrical signature to the physical properties of the examined media. They confirmed a significant positive correlation between the electrical polarization and the surface area to pore-volume ratio. Breede et al. (2012) investigated the SIP signature of sand and sand-clay mixtures (5, 10 and 20 % clay) under different saturation levels. Their results showed that the magnitude of both $\sigma'$ and $\sigma''$ were lower in sand compared to sand-clay mixtures for saturated conditions. The distinct values of $\sigma'$ and $\sigma''$ measured in channel 2 suggest that this area has different hydraulic characteristics compared to the rest of the profile. Specifically, it is consistent with the presence of a coarser textured soil, characterized by larger average grain size and lower CEC compared to the loamy soil that was measures by channels 1 and 3.

Analysis of the magnitude of the initial increase in $\sigma''$ at the different channels (between 0.5 and 1.6 hours) for the heterogeneous case showed that it was lower in the sandy layer (channel 2) compared to the loam (channels 1 and 3) by $\sim$15%. This observation can be explained by the difference in CEC between the different soil types (a difference of 17%; see section 3.1). Isomorphic exchange in the mineral structure are less substantial in sand, which in combination with its smaller specific surface area, leads to lower CEC. Hence, the changes to the Stern layer composition resulting from the $Na^+ - Ca^{2+}$ exchange were less prominent in the sand.

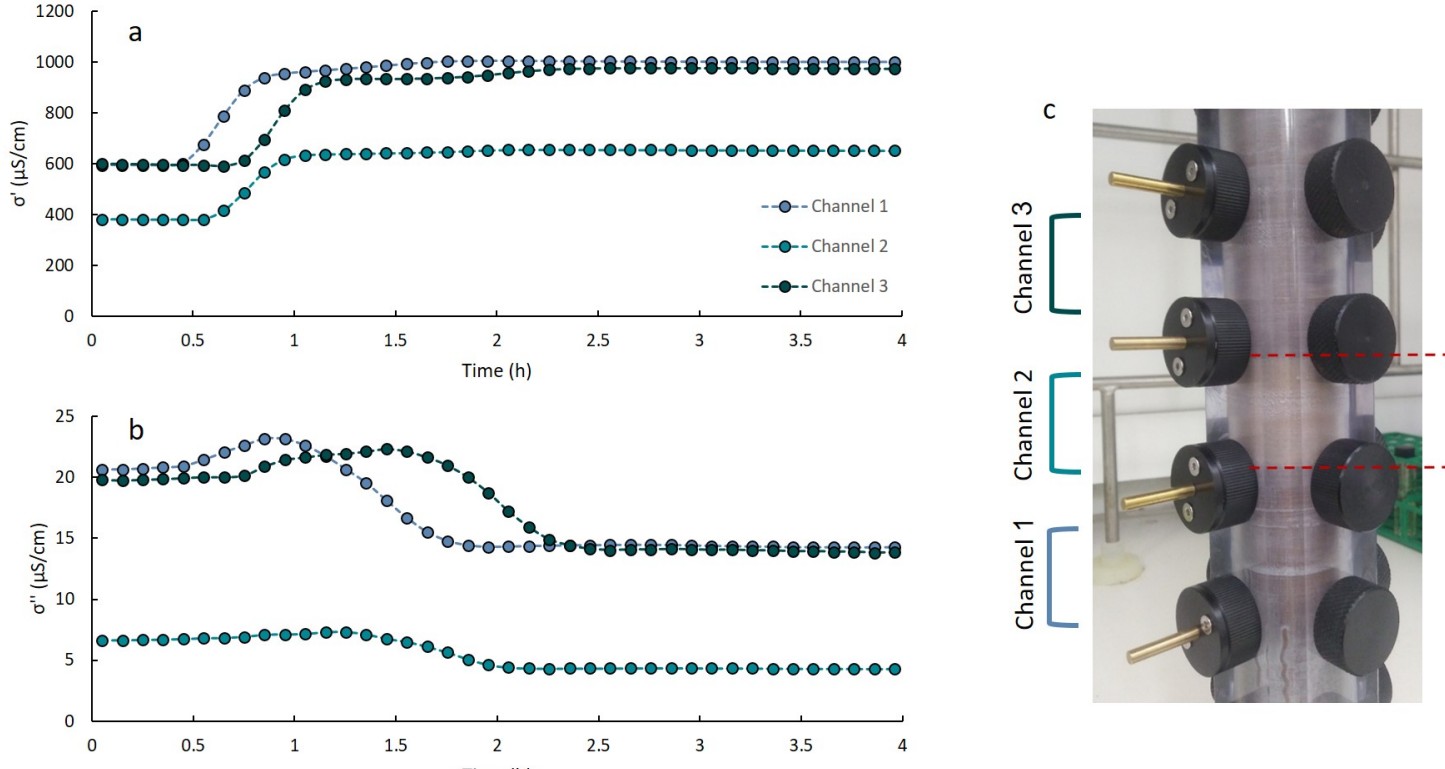

**Figure 7.** The real (a) and imaginary (b) components of the complex conductivity at 1Hz versus time, for the heterogeneous profile. (c) shows the heterogeneous profile: a sandy layer was packed between the two middle electrodes (channel 2).

## 4   Summary and conclusions

The main objective of this study was to demonstrate the ability of SIP measurements to depict BT patterns, ion exchange processes and profile heterogeneity in a simple system. The presented laboratory setup combines geo-electrical measurements backed by standard outflow EC and chemical analysis. Four experimental cases were examined: $CaCl_2$ transport through a homogeneous loamy profile, $CaCl_2$ and $ZnCl_2$ transport through a homogeneous sandy profile and $CaCl_2$ transport through a heterogeneous profile. Our results led us to the following conclusions: (1) SIP may serve as an alternative or supplementary tool for the monitoring of solute transport patterns through porous media in simple systems, requiring no outflow sampling. (2) In addition to the changes in outflow EC, the SIP measurement indicated the initiation and the end of the cation exchange along the soil column and following it, the delayed BT of the cations. (3) SIP-based BTC analysis is superior over conventional outflow analysis as it can characterize system heterogeneity, and is superior over EC-based analysis as it is capable of distinction between the adsorption end-members. Clearly, for more complex systems that involve multiple ions (either in the inflow solution or initially adsorbed to the soil) and additional processes (such as dissolution or biochemical reactions), SIP alone

may not be sufficient for full resolution of the system and some sampling and chemical analysis may be required. However, even for such systems, the use of electrical monitoring may not only allow to reduce the sampling frequency and types of analysis performed on the outflow samples, but also infer spatial heterogeneity and more complex processes than what can be achieved by outflow sampling alone. Remaining challenges include the development of appropriate methodology that combines SIP and EC/chemical-based BT analysis for complex systems, and the upscaling of this approach to field BT analysis.

*Author contributions.* PK and DE performed preliminary experimental work, SBM and AF designed the the experiments. SBM performed the experiments, analyzed the data and drafted the paper. All authors edited the paper.

*Competing interests.* The authors declare that they have no conflict of interests

*Acknowledgements.* This research was funded in part by the German-Israeli Water Technology Cooperation Program (project number WT1601/2689), the German Federal Ministry of Education and Research (BMBF) and the Israeli Ministry of Science, Technology and Space (MOST).
This research was also supported in part by the Ministry of Science & Technology, Israel & Ministero degli Affari Esteri e della Cooperazione Internazionale, Italy and also by the Israel Science Foundation (grant No. 2130/20).
The authors would like to thank the anonymous reviewers and the editor Dr. Christine Stumpp for the constructive comments that helped to improve this manuscript.

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
