# Peer review of "Geophysically-based analysis of BTCs and ion exchange processes in soil"

_Hydrology and Earth System Sciences, 2020_

## Referee Comment (RC1) · Anonymous Referee #1 · 1 Oct 2020

The manuscript presents analysis of solute transport in saturated small dimension soil columns. The paper compares breakthrough (BT) analysis that is based on standard measurement of the solution electric and chemical properties, as commonly measured on the column boundaries, with that obtained by spectral induced polarization (SIP) which is based on measurement of the variations in the bulk electrical properties of the soil column during solute transport process. The similarity between the BT obtained by both methods is not surprising and expected. Yet it reveled another possibility which is related to characterization of cation exchange processes taking place in the porous domain during the transport. Obviously, cation exchange between the solution and solid phases result in changes in bulk electrical conductivity of the domain. The analysis presented in this manuscript is very elegant and promising. Yet is hard to anticipate

broadening of methodology to applicable studies as the sensitivity of the methodology will be reduced dramatically in cases where the chemical composition of the solution is more complex and the differences in the ion concentration and transported solution is not as dramatic as in the case presented here. This subject is discussed shortly at summary. However, the paper will benefit if a broader discussion on where and how this method can be implemented and give few examples to how it can be used. There are many research areas that could benefit from better understanding flow and transport process, from remediation of contaminated sites to fertilizers distribution in the soil. Where the world is going to benefit this development with respect the of the technological limitation? Nevertheless, I recommend publication. I do not have specific comments.

---

## Referee Comment (RC2) · Anonymous Referee #2 · 12 Oct 2020

**General comments**

This manuscript presents experimental findings from simple laboratory column experiments. Spectral Induced Polarization (SIP) was measured in three column parts and signals of real and imaginary conductivity were used to describe solute transport and ion exchange in the columns. While the paper is condensed and clearly written, I am in line with the first referee that (1) novelty of the findings is rather limited and (2) that the applicability the method to real world problems in more complex systems is not proved. However, if the authors present additional data from a more complex experiment, and this is relatively easy, because they just have to fill their laboratory columns with a "real-world matrix" and use pollutants rather than salt, I see potentials for an essentail scientifc contribution.

Specific comments

(1) Novelty and research gap: SIP is an established tool and has been extensively used to investigate processes in laboratory columns. Some papers are cited in the results and discussion section, but no elaborate literature review is included in the introduction. As a result, a clear research gap is missing. The following two examples (and there are many more) exemplify the fact that the underlying processes and the different behaviour of chloride and cations in the laboratory columns have been documented by SIP before: Ion exchange in columns already 10 years ago by Vaudelet et al., WRR 2011. More recently, SIP was also used to study calcite precipitation (Izumoto et al., 2020, https://doi.org/10.1093/gji/ggz515). Thus, I agree with Referee#1 that the results of the present study are rather predictable and not surprising.

(2) Applicability to real world problems: Scientific column experiments are mainly carried out for two types of problems: (a) Solute transport in complex media (e.g. soil columns, etc.) is evaluated. Then the studied matrix inside the column is highly heterogenous and produces a complex SIP signal that can hardly be interpreted, because a variety of different factors influence the geoelectrical signature. This has been documented by various studies and is stated by the authors themselves. (b) Solute transport of specific compounds (e.g. organic pollutants, heavy metals) is studied. The transport of sodium chloride (and exchanges with calcium) are by no way representative for the transport other pollutants. For both types of problems the real value of SIP cannot be evaluated from the data presented in the present study. The only value shown in the manuscript is to evaluate transport of ions through a simple (sandy) matrix and to provide additional insights compared to EC-measurements at the column outflow. However, it is established theory that electric conductivity is a summary parameter related to the ionic strength of a solution and hence cannot be used to assess the concentration of a single ion inside a changing ion mixture.

Here the authors must present additional data that SIP is a tool for real world problems, e.g. is SIP able to detect preferential flow patterns in a heterogeneous soil matrix; or:

is SIP a tool that adds to the understanding of pollutant transport? If they do, this would increase the value of their paper and produce a piece of work worth publishing in a journal like HESS. For this, I propose that the authors fill their column with more heterogeneous media and additionally repeat their experiments also including a typical pollutant. Here heavy metals seem most promising. With their approach - fitting HYDRUS-1D to SIP-conductivities to predict breakthrough curves - they could then produce more solid knowledge on chances and limitations of SIP-data for solid transport studies in column experiments.

Technical corrections are not meaningful at this stage.

---

## Referee Comment (RC3) · Anonymous Referee #3 · 6 Nov 2020

General comments: The manuscript titled, Geophysically-based analysis of BTCs and ion exchange processes in soil, by Ben Moshe et al., details the results of continuous-injection flow-through experiments in (1) a column packed with homogenous calcareous loamy sand and (2) a loamy sand-pure sand layered column. Via a combined geo-electrical (using spectral induced polarization, SIP) and solute transport modeling approach the authors show the applicability of SIP for capturing conservative tracer (Cl-) breakthrough curves. The manuscript is generally well written and clearly organized, and provides a meaningful contribution to research focused on the joint interpretation of geophysical and geochemical datasets, specifically by coupling the two approaches with transport simulations. The quality of the experiments and data are good. The manuscript convincingly shows the added value of geo-electrical signals in capturing

solute breakthrough curves (via the real conductivity) and their potential to also capture ion-exchange processes. However, certain aspects of the joint inverse simulation considering both geo-electrical and geochemical data, such as the seemingly arbitrary consideration of only the partial real conductivity dataset need to be clearly addressed. The interpretation of the imaginary conductivity time-series is rather brief and vague. The authors should combine their time-series analysis with an interpretation of imaginary conductivity spectra to improve the mechanistic description of the ion-exchange processes driving polarization signals. In addition, the introduction and discussion would both benefit from a comprehensive review and critical consideration of the current literature. These, along with other specific comments outlined below, should be addressed before the manuscript can be considered for publication.

Specific comments: Lines 35 – 54: The introduction should clearly define what "hydrogeophysics" is. The description of polarization mechanisms does not properly distinguish between EDL-polarization and membrane polarization and this should be clarified. Several of the statements should be properly and fairly referenced. In general the introduction would benefit from a more comprehensive review of the literature. Currently, the manuscript has a very short list of references. Lines 56 – 60: Here and elsewhere, the manuscript should acknowledge previous studies that have combined breakthrough curve analysis in flow through systems with the monitoring of geo-electrical properties (e.g. Davis et al., 2006 and Deng et al., 2020) and combined these with transport models (e.g. Slater et al. 2009 and Mellage et al. 2018). How does the current work build on what has already been done? Similarly, the results of the current work should be compared to those in previous studies in the discussion. Figure 3, Results: It seems as if only a subset of the real conductivity time series data were considered in the Hydrus-1D fitting scheme, and the cutoff was different at different channels (SIP measurement locations). The authors should clearly justify what data were used for model fitting. The authors should also report fitted parameter values. I would argue that a comparably good fit could have been achieved by considering only the Cl- breakthrough. However, the authors claim that real conductivity information could replace BTC analysis. Considering the full time-series of real conductivity would likely yield a different parameter set than using only a subset of the data because of the ion exchange processes and associated signal changes later on. Thus, if only relying on SIP data for breakthrough curve analysis, most likely, a biased set of parameters would be estimated. This is contradictory to one of the main conclusions of the manuscript. Lines 196 – 200: The initial increase in the imaginary conductivity of the loam is not observed in the sand layer (in Figure 3b), however the drop in imaginary conductivity occurs in both the loam and sand layers. The authors should provide an explanation for this difference. Lines 200 – 204: Why would sodium-calcium exchange enhance ion mobility? This contradicts the interpretation of the decrease in imaginary conductivity. How would the inflowing solution and exchange process affect the interfacial/surface conductivity of the loam? SIP spectra: Currently, the manuscript presents select spectra in the supporting information. The loam exhibits a clear sharp frequency peak, indicating a dominant polarization length-scale. Fitting, for example, a Cole-Cole model to the spectra could shed light on changes in polarization length scales and or ion mobilities that would help to improve the current conceptual model provided. Figure 5c: The sand layer seems to be thin in relation to the loam layers, this is also mentioned in the text, line 210. Did the authors consider to what extent the signal in the sand-channel is influenced by the adjacent loam layers? Is the resolution of the effective electrode response volume fine enough to resolve a "sand-only" contribution? It would be good to provide a spectral comparison between the loam and the sand.

Technical comments: Line 4: change soil profiles to soil profile Line 46: What is the low frequency range? Line 65: insert "a" in – and alternating current in "a" wide range of... How was the real conductivity normalized? What frequency is plotted in the time series plots? (This could be justified by presenting spectra)

---

## Author Comment (AC1) · 10 Dec 2020

**Response to anonymous referee #1**

**We would like to thank anonymous referee #1 for the comments.
The main concern of the reviewer is related the applicability of the method to
real world problems.**

**We address this concern in the following reply.**

As the reviewer correctly pointed, we currently present a fairly simple case of solute
transport and ion-exchange. To demonstrate the applicability of SIP-based analysis
of these processes, we intend to expand our work to include reactive transport of
additional contaminants. In a series of experiments, we examined a few alternatives
(e.g ZnCl, cationic dyes). The results of these experiments qualitatively support the
findings presented in the original manuscript, but naturally add complexity by including
other processes such as precipitation

We hereby present some of the result of our additional experiments. The follow-
ing results show the real and imaginary components of the complex conductivity (at 1
Hz) over time in a sand column during a continuous Zinc injection.

The results clearly show the BT of the non-reactive ions around t=1h (the $Cl^-$ anion

and $Na^+$ cation that behaves as a non-reactive solute since the experiment started after a long $NaCl$ pre-wash). A second geo-electrical BT is observed around t=1.7h. This BT corresponds to the increased concentrations of $Ca^{2+}$ that was washed out during the Zn injection and was detected by ICP. The subsequent drop in $\sigma'$ is probably the result of Zn precipitation. PhreeqC simulations confirmed a positive precipitation potential of $Zn(OH)_2$, which is expected to be the dominant species in the system. The imaginary conductivity presented a slight increase initially and then decreased significantly (similarly to our $Ca^{2+} - Na^+$ exchange experiment). A second pattern of increase and decrease is observed between t=2-14h. This pattern corresponds (timing-wise) to the observed drop in $\sigma'$ and hence is likely to be related to the Zn precipitation process. Further laboratory work and analysis are still needed to complete this experiment. We are also currently testing similar setups for monitoring of organic pollutants and other metals.

Further, as suggested by the reviewer, the discussion will be expanded: the limitations of the method will be presented and possible applications to different areas of environmental study will be presented.

---

## Author Comment (AC3) · 10 Dec 2020

**Response to anonymous referee #2**

**We would like to thank anonymous referee #2 for the comments.
The main concerns of the reviewer are related to two aspects:**

**1.  The novelty of the presented work in light of previous research in the
field.
2. The applicability to real world problems.**

**We jointly address these concerns in the following reply.**

1. The innovation in our work (as it was presented in the original text) stems primarily
from the heterogeneous experiment.  To the best of our knowledge, this aspect was
not examined before. In the text, we discuss our results in light of previous studies that
looked into the SIP response of different-textured soil in no-flow conditions.  We also
suggest a connection between the imaginary SIP response (in the sand compared to
the loam) and the CEC of the different soils. In the revised version of the paper we will
support our findings with CEC measurements of the sand and loam.
We agree that expanding the paper (by additional experiments with other contami-
nants) would enhance the novelty of our work, in addition to addressing the second
concern. We thus intend to do so (as elaborated in section 2).

2. The transport of additional solutes will be examined and presented. In a series of experiments, we examined a few alternatives.

We hereby present some of the result of our additional experiments. The following results show the real and imaginary components of the complex conductivity (at 1 Hz) over time in a sand column during a continuous Zinc injection.

The results clearly show the BT of the non-reactive ions around t=1h (the $Cl^-$ anion and $Na^+$ cation that behaves as a non-reactive solute since the experiment started after a long $NaCl$ pre-wash). A second geo-electrical BT is observed around t=1.7h. This BT corresponds to the increased concentrations of $Ca^{2+}$ that was washed out during the Zn injection and was detected by ICP. The subsequent drop in $\sigma'$ is probably the result of Zn precipitation. PhreeqC simulations confirmed a positive precipitation potential of $Zn(OH)_2$, which is expected to be the dominant species in the system.

The imaginary conductivity presented a slight increase initially and then decreased significantly (similarly to our $Ca^{2+} - Na^+$ exchange experiment). A second pattern of increase and decrease is observed between t=2-14h. This pattern corresponds (timing-wise) to the observed drop in $\sigma'$ and hence is likely to be related to the Zn precipitation process. Further laboratory work and analysis are still needed to complete this experiment. We are also currently testing similar setups for monitoring of organic pollutants and other metals.

3. The introduction will be expanded to include additional papers that addressed the link between BTCs / ion-exchange processes and SIP monitoring in soil. For example, Izumoto et al., 2020, Slater et al., 2009 and Mellage et al., 2018 will be included as examples of previous studies that combined geo-electrical monitoring with transport processes or ADE modeling.

References:

1. Izumoto, S., Huisman, J. A., Wu, Y., and Vereecken, H. (2020). Effect of solute concentration on the spectral induced polarization response of calcite precipitation. Geophysical Journal International, 220(2), 1187-1196.‏
2. Slater, L. D., Day‐Lewis, F. D., Ntarlagiannis, D., O'Brien, M., and Yee, N. (2009). Geoelectrical measurement and modeling of biogeochemical breakthrough behavior during microbial activity. Geophysical research letters, 36(14).‏
3. Mellage, A., Holmes, A. B., Linley, S., Vallee, L., Rezanezhad, F., Thomson, N.,and Van Cappellen, P. (2018). Sensing coated iron-oxide nanoparticles with spectral induced polarization (SIP): experiments in natural sand packed flow-through columns. Environmental science and technology, 52(24), 14256-14265.‏
Interactive
comment

---

## Author Comment (AC4) · 10 Dec 2020

**Response to anonymous referee #3**

**We would like to thank anonymous referee #3 for the comments. We will account for them in a revised version of the paper, as we report in the following point–by–point reply:**

**General comments (GC)**

**GC 1 -** *The seemingly arbitrary consideration of only the partial real conductivity dataset need to be clearly addressed*

**Autors' response -** The complete real conductivity signal (over the entire experiment) reflects the combination of the changes in pore-electrolyte conductivity due to the different processes that occur in the system over this time. Since the ADE modeling presented in the text reflects only the transport of the non-reactive ions, only the data of the initial BT was considered. We accept that this needs to be specifically clarified and will account for that in the revised version. Additionally, we intend to improve the ADE-modeling part of the paper by (a) simulating the non-reactive ion transport not only for $Cl^-$ but as a combination of all non-reactive species (in our system $Na^+$ and a fraction of the $Ca^{2+}$ ions that behave as non-reactive solutes) and (b) present a model fit and prediction for the $Ca^{2+}$ ions based on the SIP data of the
secondary BT (t=0.9-2.5h). Transport parameters were originally included in SI and will now be presented in the main text body.

While the reviewer is correct that calibrating a model using SIP data would yield in a different set of parameters (compared to 'out-of-column' calibration). Yet, one may argue that with proper geomerical considerations the SIP-based calibration is superior (as it contains more information). Our intention is not necesarily to replace conventional calibration but to enhance it using more spatial data and exchange-related data

**GC 2 -** *The interpretation of the imaginary conductivity time-series is rather brief and vague. The authors should combine their time-series analysis with an interpretation of imaginary conductivity spectra to improve the mechanistic description of the ion-exchange processes driving polarization signals.*

**Autors' response -** We are not sure that we understand this comment properly. The imaginary conductivity results are discussed at length (starting from line 182). We directly connect the observed drop in the imaginary conductivity to the $Ca^{2+} - Na^+$ exchange process and the ions' mobility. However, we agree that the discussion around the initial increase in $\sigma''$ (before it decreases) is not wide enough. It is worth noting in this context, that while similar patterns had been reported before (see Vaudelet et al., 2011), no explanation was suggested. We intend to expand the discussion around this part.

**GC 3 -** *the introduction and discussion would both benefit from a comprehensive review and critical consideration of the current literature.*

**Autors' response -** We fully accept this comment. The literature review included in the introduction will be expanded. For example, Izumoto et al., 2020, Slater et al., 2009 and Mellage et al., 2018 will be included as examples of previous studies that combined geo-electrical monitoring with transport processes or ADE modeling.

**Specific comments (SC)**

**SC 1 -** *Lines 35 – 54: The introduction should clearly define what "hydrogeo-physics" is. The description of polarization mechanisms does not properly distinguish between EDL-polarization and membrane polarization and this should be clarified. Several of the statements should be properly and fairly referenced. In general the introduction would benefit from a more comprehensive review of the literature. Currently, the manuscript has a very short list of references.*

**Autors' response -** We fully accept this comment. The introduction section of the revised version will include (as suggested by the reviewer) a definition of the term "hydrogeophysics" as well as a clear description of both EDL and the membrane polarization mechanisms and their contribution to the observed signal in soil systems. Additionally, we intend to significantly expend the literature review in the introduction and the discussion (see GC3).

**SC 2 -** *Lines 56 – 60: Here and elsewhere, the manuscript should acknowledge previous studies that have combined breakthrough curve analysis in flow through systems with the monitoring of geoelectrical properties (e.g. Davis et al., 2006 and Deng et al., 2020) and combined these with transport models (e.g. Slater et al. 2009 and Mellage et al. 2018). How does the current work build on what has already been done? Similarly, the results of the current work should be compared to those in previous studies in the discussion.*

**Autors' response -** We fully accept this comment and will include proper reference to those (and other) works (see response to GC3).

[Figure]

**SC 3 -** *Figure 3, Results: It seems as if only a subset of the real conductivity time series data were considered in the Hydrus-1D fitting scheme, and the cutoff was different at different channels (SIP measurement locations). The authors should clearly justify what data were used for model fitting. The authors should also report fitted parameter values. I would argue that a comparably good fit could have been achieved by considering only the Cl- breakthrough. However, the authors claim that real conductivity information could replace BTC analysis. Considering the full time-series of real conductivity would likely yield a different parameter set than using only a subset of the data because of the ion exchange processes and associated signal changes later on. Thus, if only relying on SIP data for breakthrough curve analysis, most likely, a biased set of parameters would be estimated. This is contradictory to one of the main conclusions of the manuscript.*

**Autors' response -** This comment is addressed in the 'General comments' section. Please see response to SC1.

**SC 4 -** *Lines 196 – 200: The initial increase in the imaginary conductivity of the loam is not observed in the sand layer (in Figure 3b), however the drop in imaginary conductivity occurs in both the loam and sand layers. The authors should provide an explanation for this difference.*

**Autors' response -** The initial increase in imaginary conductivity does occur for the sand layer as well (we assume that the reviewer intended to refer to Fig. 5b). This can be seen clearly in the attached figure below, which is an enlarged version of the sandy layer $\sigma''$ vs time graph.

   **SC 5 -** *Lines 200 – 204: Why would sodium-calcium exchange enhance ion mobility? This contradicts the interpretation of the decrease in imaginary conductivity. How would the inflowing solution and exchange process affect the interfacial/surface*

*conductivity of the loam?*

**Autors' response -** As presented in the methods section (SIP subsection), the complex conductivity is related to ion mobility in the EDL through the mobility, denoted as $\beta$. Imaginary conductivity changes have been shown to be related to the mobility of ions in the stern layer (e.g. Leroy et al, 2009, Vaudelet et al, 2011, Schwartz et al, 2012). As we explain in the text (line 186): "Adsorbed $Na^+$ ions maintain their hydration shell and hence, are weakly adsorbed to the soil and are more mobile compared to $Ca^{2+}$. During the $Na^+ - Ca^{2+}$ exchange, the less mobile $Ca^{2+}$ ions occupied the stern layer and caused the decrease in $\sigma''$."

**SC 6 -** *SIP spectra: Currently, the manuscript presents select spectra in the supporting information. The loam exhibits a clear sharp frequency peak, indicating a dominant polarization length-scale. Fitting, for example, a Cole-Cole model to the spectra could shed light on changes in polarization length scales and or ion mobilities that would help to improve the current conceptual model provided.*

**Autors' response -** We accept this comment. The revised version of the manuscript will include a Cole-Cole model fit for obtained spectra to demonstrate the change in the polarization length scale as a result of the $Ca^{2+} - Na^+$ exchange.

**SC 7 -** *Figure 5c: The sand layer seems to be thin in relation to the loam layers, this is also mentioned in the text, line 210. Did the authors consider to what extent the signal in the sand-channel is influenced by the adjacent loam layers? Is the resolution of the effective electrode response volume fine enough to resolve a "sand-only" contribution? It would be good to provide a spectral comparison between the loam and the sand.*

**Autors' response -** We agree that the electrodes measuring over the sandy

part of the profile don't measure exclusively sand. For this reason, we refer to this layer as 'the sandy layer' and don't use the term 'sand only'. However, in our experimental setup the pairs of electrodes are fairly spaced and it is reasonable to assume that the sand layer is dominant in the signature captured by these electrodes. While this was not tested to the current geometry, it is clear (see for example Furman et al., 2003) that the region between the potential electrodes dominates the signal.

**Technical comments**

1. Line 4: change soil profiles to soil profile

**Autors' response -** Thank you for the attention. The sentence was corrected and now reads: "In this work...in homogeneous and heterogeneous soil profiles" (i.e the word 'a' was removed).

2. Line 46: What is the low frequency range?

**Autors' response -** We refer to the range of 1 mHz to 1kHz as the low frequency range characteristic to IP. This is now specifically mentioned in the text.

3. Line 65: insert "a" in – and alternating current in "a" wide range of...

**Autors' response -** Corrected according to the comment.

4. How was the real conductivity normalized?

**Autors' response -** The real conductivity was normalized to a 0-1 scale according to the following equation:

$$\sigma'_{norm} = \frac{\sigma' - \sigma'_{min}}{\sigma'_{max} - \sigma'_{min}} \qquad (1)$$

5. What frequency is plotted in the time series plots? (This could be justified by presenting spectra)

**Autors' response -** All SIP time series figures are for a frequency of 1Hz. This is clearly mentioned in line 149 and in the captions of the figures.

References:

1. Izumoto, S., Huisman, J. A., Wu, Y., and Vereecken, H. (2020). Effect of solute concentration on the spectral induced polarization response of calcite precipitation. Geophysical Journal International, 220(2), 1187-1196.‏
2. Slater, L. D., Day‐Lewis, F. D., Ntarlagiannis, D., O'Brien, M., and Yee, N. (2009). Geoelectrical measurement and modeling of biogeochemical breakthrough behavior during microbial activity. Geophysical research letters, 36(14).‏
3. Mellage, A., Holmes, A. B., Linley, S., Vallee, L., Rezanezhad, F., Thomson, N.,and Van Cappellen, P. (2018). Sensing coated iron-oxide nanoparticles with spectral induced polarization (SIP): experiments in natural sand packed flow-through columns. Environmental science and technology, 52(24), 14256-14265.‏
4. Leroy, P., and Revil, A. (2009). A mechanistic model for the spectral induced polarization of clay materials. Journal of Geophysical Research: Solid Earth, 114(B10).‏
5. Schwartz, N., Huisman, J. A., and Furman, A. (2012). The effect of NAPL on the electrical properties of unsaturated porous media. Geophysical Journal International, 188(3), 1007-1011.
‏ 6. Vaudelet, P., Revil, A., Schmutz, M., Franceschi, M., and Bégassat, P. (2011). Induced polarization signatures of catins exhibiting differential sorption behaviors in

saturated sands. Water Resources Research, 47(2).

7. Furman, A., Ferré, T. P., and Warrick, A. W. (2003). A sensitivity analysis of electrical resistivity tomography array types using analytical element modeling. Vadose Zone Journal, 2(3), 416-423.‏‏ ‏

---

## Author Response (AR2)

**Response to review - Geophysically-based analysis of BTCs and ion exchange processes in soil (Ben Moshe et al.)**

**Dear Dr. Stumpp,**
**We thank you and the two anonymous referees for the comments. We find all of the comments constructive and account for them in a revised version of the paper, as we report in the following reply:**

**Referee #1**

**Specific comments**

1. *The description of fitting the "non-reactive" vs "reactive" cases is confusing. Figure 4 presents the full time series SIP dataset based on the cumulative experimental time, but figures 4b and c seem to redefine the starting time. It would be clearer if everything was plotted in relation to a single start time, but the injection of the different solutions was clearly highlighted.*
**Author response** - We believe that the referee misunderstood the description of the experiment represented by Fig.4. Figures 4a-c have the same starting time. Each experiment started with a 'pre-wash' stage in which the column was flushed with a background solution of $NaCl$. After the outflow EC stabilizes on the same value of the background solution, it was replaced by the inflow solution (of either $CaCl_2$ or $ZnCl_2$). The replacement of the background solution by the $CaCl_2$ or $ZnCl_2$ solutions marked $t = 0$. Since this was not clear we now state this specifically in the description of the column experiments (see Line 122-125 in the revised text).

2. *Equations 6 and 7 imply that the authors assumed linear retardation for the Ca2+ and Zn2+ ions. Is this justified for this particular loamy soil? Is there evidence to suggest that Ca and Zn sorption could exhibit a saturation-type (Langmuir/Freundlich) behavior?*
**Author response** - Linear adsorption patterns is assumed here for simplicity. Obviously, a non-linear (Langmuir or Freundlich-type isotherms) could have been used for the calibration. Since the values of the adsorption coefficient were not the main concern here (rather the demonstration of the model calibration using the SIP data) we see it as a reasonable approximation given the relatively narrow range of concentrations considered in this study.

3. *Are the estimated parameters from the SIP-based calibration able to capture the Na+ and Ca2+ breakthrough behavior when considering that both ions are present at the same time? (Figure 3b)*
**Author response** - The simple model we present here divides the SIP data into two parts (reactive and non-reactive species) and the calibration of each part is performed separately. It is correct that at this point we use only real conductivity for both the reactive and non-reactive model. To model the full SIP data, a more complex model (that includes the transport of the three species involved ($Na^+$, $Ca^{2+}$ and $Cl^-$) and the exchange reaction between the cations) should be constructed. However, the aim of this work is limited to demonstration of the sensitivity of SIP to the BT of the different ions in the system and to the presentation of a simple and

efficient way to predict the progression of both the dissolved species and the exchange process.

4. *What subset of the Ca2+ breakthrough concentrations is plotted in Figure 4c, and how does that differ from what is shown in Figure 3b?*
**Author response** - The outflow $Ca^{2+}$ concentrations in Fig.4c represent the fraction of the $Ca^{2+}$ ions that participated in the exchange reaction (see explanation in line 184 in the revised version). In light of this comment we added an additional note in lines 214-215 of the revised version and modified the caption of the figure accordingly.

5. *Lines 224 – 225: I find the distinction between a reactive Ca2+ fraction and a non-reactive fraction misleading. An alternative, process-based approach, would be to compare the imaginary conductivity profiles to the adsorbed reactive species concentrations and attempt to calibrate the model that way, in turn validating it with the concentration breakthrough curve. For the latter case perhaps using the imaginary conductivity data presented in 6b would be best.*
**Author response** - We agree that such non-orthodox separation may be confusing. We have indeed considered this option. The role of $\sigma''$ as an indicator of the $Na^+ - Ca^{2+}$ exchange progression suggests that the $\sigma''$ values can serve as a proxy for the adsorbed concentration of the retarded (reactive) species in an alternative modeling approach. In light of this comment, we now discuss this modeling approach in the paper (see lines 261-266 in the revised version). We also present in the supplementary material a demonstration of this approach. We normalized the $\sigma''$ data and used it as a proxy for the adsorbed concentration in a HYDRUS simulation. Our main concern regarding this approach is that in order to accurately model the initial increase in $\sigma''$ further investigation and modeling effort are needed, and mostly the inclusion of non-equilibrium component in the model. While this is feasible, as the supplementary information suggests), we think that including that in the main text will be somewhat confusing, and more importantly, divert the focus of the manuscript from BTC to non-equilibrium exchange.

6. *What non-reactive species concentration is being presented in Figure 4b: Cl-, Na+?*
**Author response** - As explained in lines 184 and 214-215, the non-reactive species in Fig. 4b are $Cl_-$, $Na^+$ and the fraction of the $Ca^{2+}$ ions that did not participate in the exchange reaction (i.e., the fraction of the $Ca^{2+}$ ions that reached the outlet of the column together with the non-reactive species).

7. *Lines 269-280: Yes, but the SIP channels are at "earlier" locations within the columns. To better address this interpretation perhaps a comparison between imaginary conductivity and Cl-concentrations at those locations in the columns would be warranted. I find the use of "stability" vague here. Perhaps the authors could elaborate. An alternative explanation could be the interplay between the contributions of Stern- and Diffuse-layer polarization during the exchange process. The Na-Ca exchange may have released higher-mobility Na ions that temporarily remained in the diffuse layer (contributing to diffuse layer polarization) before moving into the bulk solution.*
**Author response** - Yes, the reviewer is suggesting in different words the same mechanism that we tried to suggest. The term "instability" is perhaps somewhat misleading and we removed it from the revised version. During the exchange process, ions that leave the stern layer are "weaker" in their connection to the soil grains and therefore are associated with higher mobility. At the same time, ions that are "on their way in" are still more mobile than they would be once they complete their journey to the Stern layer. Considering simple EDL model, this indeed suggest that ions are still at the diffuse layer, as the reviewer suggests. To address this point we have rephrased the relevant text in lines 252-255 in the revised text.

8. *Lines 297 – 300: Why not fit the "reactive" model form Zn and Ca using the imaginary*

*conductivity (as the imaginary resistivity)? Figure S2 again shows that only a subset of the real conductivity dataset was used for the calibration of Ca and Zn transport, this is poorly described and also not justified.*

**Author response** - Thanks. This point is already discussed and answered in our response to comment 5, above, including a demonstration (in the supplementary materials) of the way such an approach can be used.

**Minor comments**

1. *Presenting things either in PV or time makes interpreting the results confusing. Please choose one standard.*
**Author response** - Fig. 3 is the only relevant figure that was presented in terms of pore-volumes. To make it clearer, we changed it to time-based but added, as a secondary axis, the pore-volume presentation.

2. *Figure 2: I find the y-axis location at 1 Hz an odd way to present the data, it also partly covers the numerical values on the y-axis. I recommend moving the axis to 0.1 Hz (the left hand side limit).*
**Author response** - Corrected according to the comment.

3. *In general, I find the manuscript has too many acronyms, and the use of BT vs BTC is also confusing. In addition, I recommend not to abbreviate geo-electrical with GE.*
**Author response** - We respectfully disagree and choose to keep the use of acronyms where appropriate. We do agree however to spell-out GE as it may be wrongly interpreted

4. *"Stern layer" should be capitalized*
**Author response** - Corrected according to the comment

5. *Line 4: "soil profiles" to "soil profile"*
**Author response** - Corrected according to the comment

6. *Line 5: "The SIP signature was recorded..."*
**Author response** - Corrected according to the comment

7. *Line 9 and 11: A conductivity does not "react". Recommend to change to "changed in response to"*
**Author response** - Corrected according to the comment

8. *Line 139: "denoted"*
**Author response** - Corrected according to the comment

9. *Line 145: "Stokes" law*
**Author response** - Corrected according to the comment

10. *Line 152: inductively coupled plasma - optical emission spectrometer*
**Author response** - Corrected according to the comment

11. *Line 224: "constructed"*
**Author response** - Corrected according to the comment

12. *Line 235: 4b and 4b?*
**Author response** - Corrected according to the comment

**Referee #2**

*Overall, the improvements of the paper are acknowledged. As a remaining minor point it is suggested to use the same loamy matrix which has been used to document Na-Ca exchanges also for the Zinc experiment. This would add to the overall logic of the paper (only one homogeneous matrix is used) and probably also emphasize differences between Ca and Zn.*
**Author response** - Thanks. While we agree that adding an experiment with $Zn$ and the loamy soil would make the paper more complete, we do not really see how the content (i.e., the main concepts conveyed in the manuscript) would benefit from such addition. Our approach is to be as concise as possible and therefore we choose not to include this additional data.